# Detection of Spatial and Temporal Patterns of Liana Infestation Using Satellite-Derived Imagery

Chris J. Chandler *, Geertje M. F. van der Heijden, Doreen S. Boyd [ID] and Giles M. Foody [ID]

School of Geography, University of Nottingham, University Park, Nottingham NG7 2RD, UK;
Geertje.Vanderheijden@nottingham.ac.uk (G.M.F.v.d.H.); doreen.boyd@nottingham.ac.uk (D.S.B.);
giles.foody@nottingham.ac.uk (G.M.F.)
* Correspondence: Christopher.chandler1@nottingham.ac.uk

**Abstract:** Lianas (woody vines) play a key role in tropical forest dynamics because of their strong influence on tree growth, mortality and regeneration. Assessing liana infestation over large areas is critical to understand the factors that drive their spatial distribution and to monitor change over time. However, it currently remains unclear whether satellite-based imagery can be used to detect liana infestation across closed-canopy forests and therefore if satellite-observed changes in liana infestation can be detected over time and in response to climatic conditions. Here, we aim to determine the efficacy of satellite-based remote sensing for the detection of spatial and temporal patterns of liana infestation across a primary and selectively logged aseasonal forest in Sabah, Borneo. We used predicted liana infestation derived from airborne hyperspectral data to train a neural network classification for prediction across four Sentinel-2 satellite-based images from 2016 to 2019. Our results showed that liana infestation was positively related to an increase in Greenness Index (GI), a simple metric relating to the amount of photosynthetically active green leaves. Furthermore, this relationship was observed in different forest types and during (2016), as well as after (2017–2019), an El Niño-induced drought. Using a neural network classification, we assessed liana infestation over time and showed an increase in the percentage of severely (>75%) liana infested pixels from 12.9% $\pm$ 0.63 (95% CI) in 2016 to 17.3% $\pm$ 2 in 2019. This implies that reports of increasing liana abundance may be more wide-spread than currently assumed. This is the first study to show that liana infestation can be accurately detected across closed-canopy tropical forests using satellite-based imagery. Furthermore, the detection of liana infestation during both dry and wet years and across forest types suggests this method should be broadly applicable across tropical forests. This work therefore advances our ability to explore the drivers responsible for patterns of liana infestation at multiple spatial and temporal scales and to quantify liana-induced impacts on carbon dynamics in tropical forests globally.

**Keywords:** airborne hyperspectral and LiDAR; aseasonal forest; Greenness Index; liana infestation; Sentinel-2 imagery



## 1. Introduction

Lianas (woody vines) are a pervasive component of tropical forests [1,2]. They are non-self-supporting structural parasites that use the architecture of trees to extend their leaves to the forest canopy [3]. As competition between lianas and trees is stronger than tree-tree competition [4], lianas can negatively impact the growth [5,6] and survival of their host [7,8] and therefore suppress the ability of tropical forests to sequester and store carbon [9].

Lianas have been proliferating in some tropical forests [10,11], which may lead to a stronger negative impact on carbon storage and sequestration in these areas. Several putative mechanisms have been suggested for this increase, such as elevated atmospheric CO2, an increase in forest disturbance and an increase in the frequency and severity of

droughts [11]. However, it is currently still unknown which driver(s) may be responsible for changes in liana biomass and abundance over time. Additionally, while there is compelling evidence that lianas are increasing in many Neotropical forests [10,12], this may not be a global phenomenon [13]. This suggests that liana proliferation over time may be driven by regional rather than global drivers. However, in order to provide insights into the factors responsible for changes in liana abundance and to test whether these differ geographically, wide-spread monitoring of lianas over time and across large areas is essential.

Most previous studies which have assessed temporal changes in liana abundance, biomass or infestation have been based on ground data collected from permanent sampling plots [10,14,15]. However, while field-based studies may not be limited in their geographical extent, they are limited by the total area that can be feasibly sampled. This may be particularly problematic if plot-based research is unable to capture sufficient variation in environmental variables through space and time to disentangle the driving forces behind change [16].

Remote sensing technologies may provide a solution to extend field-based knowledge to larger spatial and temporal scales. However, they are dependent on the ability to detect liana infestation. Many studies have shown that lianas, as a plant group, can be distinguished from trees based on their spectral reflectance, particularly in the visible (400–690 nm) and Near Infrared (NIR)-region (700–1340 nm) [17–21], as well as thermal properties [22,23]. Subsequently, recent research has successfully detected lianas using data acquired from; UAVs, fitted with RGB [24,25] and thermal [26] sensors, satellite imagery [27] and airborne hyperspectral imagery in seasonal [28] and aseasonal forests [29]. While airborne sensors have the potential to provide high spatial and spectral resolution imagery which can be used to detect liana infestation at landscape-scales, satellite-based sensors can typically afford more frequent measurements across much larger geographical extents. However, there are a number of limitations which may pose challenges for assessing liana infestation with satellite-based remote sensing.

Firstly, spectral reflectance derived from multispectral satellites can be limited in scope as data represent non-contiguous regions of the light spectrum. Thus, a single value for each band is associated with the spectral reflectance from large regions of the spectrum [30]. Crucially, however, some bands cover smaller regions than others and may align with areas of the spectrum that are important for the discrimination of lianas and trees. In turn, this may limit the accurate detection of liana infestation to specific satellite sensors, which have spectral bands that represent similar regions of the spectrum. For example, research by Foster et al. (2008) assessed the spatial distribution of liana infestation in large canopy gaps using satellite-based hyperspectral imagery (EO-1 Hyperion: 220 10 nm bands covering 400–2500 nm). However, while Hyperion imagery was used to detect liana-dominated patches for training purposes, the prediction of liana infestation across Landsat imagery was achieved by using minimum values of brightness and greenness. The use of a simple vegetation index, such as greenness, which relates to the amount of photosynthetically active green leaves, is attractive for its ability to transfer across different sensors. However, as the study by Foster et al. (2008) was conducted in the dry season and the detection of lianas was limited to severely liana-dominated patches, it remains unclear whether multispectral satellite-based imagery, or a simple vegetation index, could be used to successfully detect liana infestation across a dense, closed-canopy aseasonal forest.

Secondly, variation in spectral reflectance between forest types may restrict the detection of lianas over broad geographical-scales if the difference is greater than that of trees and lianas. For example, logged forests typically have higher spectral reflectance compared to primary forests [31,32]. Differences in spectral reflectance may be driven by the fact that the canopies of logged forests are typically more homogenous whereas those of primary forests contain a mix of tree sizes and multiple canopy layers [33,34]. This greater structural heterogeneity can result in an increase in tree shadow and a decrease in overall reflectance in primary forests [31] which in turn may affect predictions of liana infestation [35].



Thirdly, liana chemistry tends to converge with that of trees in aseasonal tropical forests or those with high annual precipitation [36] and therefore reflectance spectra for lianas and trees are not as clearly separable [17,37]. Higher reflectance of liana leaves has been shown to be strongly related to the level of chlorophyll content [17], which is known to be more similar in wet conditions or within aseasonal forests [37]. As a result, differences in satellite-derived spectral reflectance between lianas and trees is likely to be more difficult in aseasonal forests, particularly if spectral resolution is limited to relatively few wavebands. It is therefore essential to test whether lianas can be detected in aseasonal tropical forests using satellite-based remote sensing in order to advance our ability to assess the spatial distribution of liana infestation globally.

Here we therefore aim to determine the efficacy of satellite-based remote sensing for the detection of liana infestation across an aseasonal tropical forest in Sabah, Borneo. Additionally, we assess the detectability within primary and selectively logged forests as well as during and after a period of El Niño-induced drought. We therefore aim to test whether (1) a response to drought facilitates the differentiation in spectral reflectance for lianas versus trees (Q1), (2) one single vegetation index is capable of predicting liana infestation (Q2), (3) liana infestation can be detected in satellite-based imagery using a neural network classification trained by airborne-derived liana infestation (Q3) and (4) temporal changes in liana infestation can be observed using a time-series of satellite-based imagery (Q4).

## 2. Materials and Methods

### 2.1. Study Area

This study was based in an aseasonal tropical forest in Danum Valley, Malaysia which contains a mix of primary and selectively logged lowland Dipterocarp forest (Figure 1). The Danum Valley Conservation Area (DVCA) represents a large swathe (438 km$^2$) of intact primary tropical forest. The area surrounding the DVCA has been selectively logged at varying intensities between 1972 and 1993 [38]. The vegetation within the primary forest is dominated by Dipterocarps [39], whereas the logged forest has received targeted removal of larger Dipterocarps and now has a higher proportion of fast-growing, early successional species [40]. The climate is typical of the aseasonal tropics with a mean annual temperature of 26.7 °C and an average yearly rainfall of 2900 mm [40]. Daily rainfall, temperature and solar radiation have been recorded at the Danum Valley Field Centre and are freely available for download from the South-East Asian Rainforest Research Partnership (www.searrp.org/scientists/available-data/). Borneo has one of the most aseasonal climates of any tropical region [41], although droughts do occur infrequently and usually in association with an El Niño event [42,43]. The most recent El Niño occurred in 2015–2016, peaking in early 2016 [44].

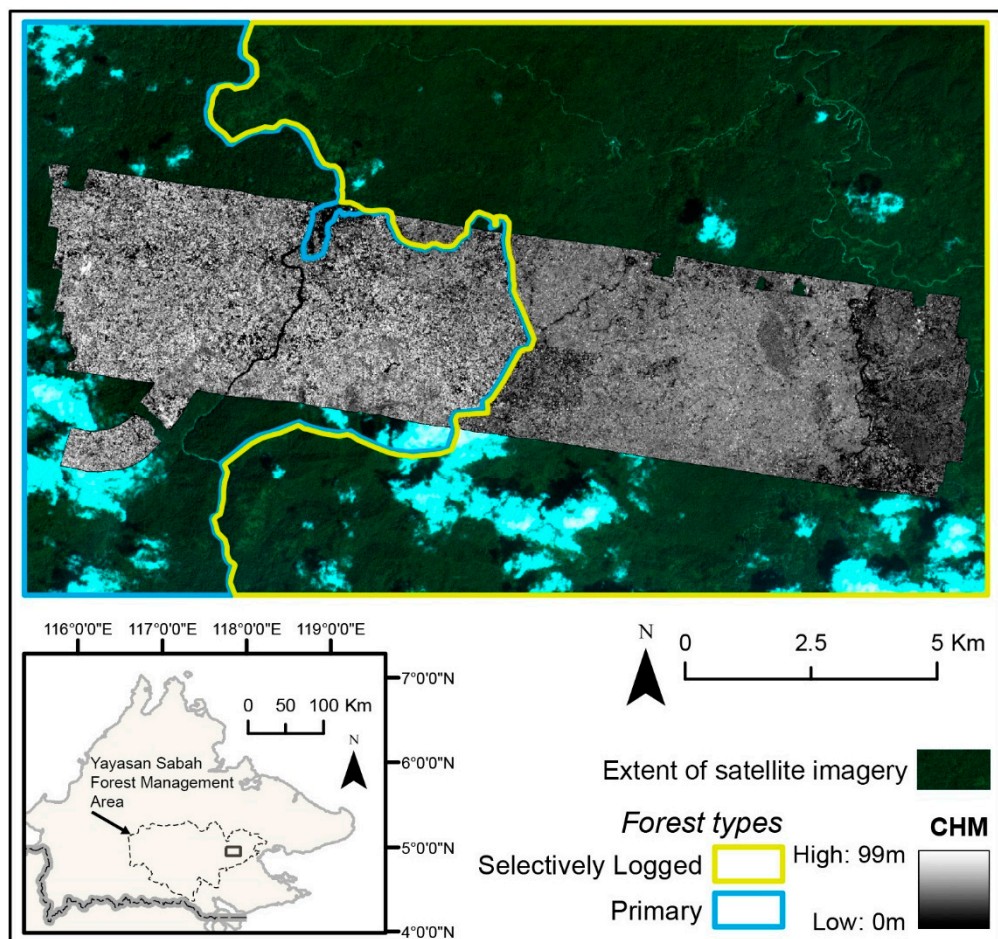

**Figure 1.** Location of the study area in Sabah, Malaysia, showing the extent of the air- and space-borne remotely sensed imagery across a primary and selectively logged tropical forest.

### 2.2. Airborne-Derived Liana Infestation Assessment

#### 2.2.1. Airborne Hyperspectral and LiDAR Data

Hyperspectral and LiDAR data were captured in November 2014 by the Natural Environmental Research Council (NERC) Airborne Research Facility (ARF). An inbuilt AisaFENIX sensor (Specim Spectral Imaging, Finland) was used to capture hyperspectral data from the visible to short wave infrared (380–2500 nm) with a spatial resolution of 3 m. A Leica ALS50-II system was used to capture both Discrete (DR) and Full Waveform (FW) airborne laser scanning (ALS) data with a spatial resolution of 1m (see Chandler et al. (2021) for full details on the airborne data collection).

#### 2.2.2. Field Data Collection

We collected liana canopy cover data in the field over a three-year time period (2017–2019). We identified individual tree crowns that were fully exposed from above using a tablet computer connected to a GPS with a Canopy Height Model (CHM) preloaded. Tree crowns were then visually assessed, by a minimum of two people, to estimate the degree of liana infestation to the nearest 5%. Each estimate was discussed and a final estimate mutually agreed [28]. Tree crowns were then manually delineated on the CHM using the GeoEditor application (MapTiler).

#### 2.2.3. Modelling Liana Infestation

We delineated a total of 724 trees with liana canopy cover estimates ranging from 0 to 100%, which corresponded to 21,822 pixels from the hyperspectral imagery that were fully

inside tree crown boundaries. Airborne hyperspectral data were used to train a neural network classification to predict liana infestation across the full extent of the airborne survey. Neural networks, in contrast with many other prediction techniques, can learn hidden relationships without imposing restrictions on the data. We used a multi-layer perceptron (MLP) network with resilient backpropagation and weight backtracking, thus parameters such as learning rate and momentum are not required. The model architecture consisted of an input layer with eight principal components, one hidden layer with four units (neurons) and an output layer with two units corresponding to either a tree or liana class. A sigmoid activation function was applied to the hidden units and therefore the output values were restricted to a range between 0 and 1. The output from the neural network represents a measure of the strength of class membership which can be used to derive a soft classification relating to the proportion of liana infestation [45,46]. We accounted for error in liana canopy cover estimates which may have changed during the time lag (2.5–3.5 years) between airborne data acquisition and the ground survey of liana canopy cover estimates. Following the methodology used in Chandler et al. (2021), we randomly reclassified 5% of all pixels with no liana infestation and classified them as severely liana infested. Similarly, we selected 11% of pixels with severe liana infestation and classified them as liana-free. Even a large degree of error in the training data (>30%) has shown to not impact the ability of the model to predict liana infestation with good accuracy [29]. We ran the neural network model 100 times and after each iteration the model was applied to the entire study landscape. The average of the 100 neural network outputs was used to produce a final landscape scale liana infestation map.

### 2.3. Satellite-Derived Liana Infestation Assessment

### 2.3.1. Satellite-Based Data

We used freely available bottom of atmosphere reflectance Sentinel-2 imagery downloaded from the United States Geological Survey (USGS) Earth Explorer (https://earthexplorer.usgs.gov/ (accessed on 5 June 2021)). The earliest image with limited cloud cover to use in combination with airborne imagery (2014), was obtained in May 2016. This also aligned with the end of an El Niño-induced drought period in which there were higher temperatures [47] and a significant reduction in precipitation between November 2015 and April 2016 in Danum Valley Conservation Area (DVCA) [44]. We collected additional imagery in approximately one-year time intervals, depending on when cloud-free images could be obtained (i.e., November 2017, June 2018 and April 2019). Areas contaminated by cloud and cloud shadow were manually delineated and removed from each image. As the spatial resolution of Sentinel-2 bands range from 10 m to 60 m we resampled all bands to a spatial resolution of 10m so they could be aligned at the same scale. These images were then used to produce a time series in order to assess whether changes in liana infestation can be observed over time.

### 2.3.2. Spectral Reflectance for Lianas versus Trees

To identify which spectral bands from Sentinel-2 imagery may be most important for discriminating lianas from trees, we assessed the difference in spectra derived from airborne hyperspectral imagery for liana-free and liana-infested (>75%) trees specifically within the Sentinel-2 spectral band regions. We calculated the average difference in reflectance between the two infestation classes for all hyperspectral bands that aligned with Sentinel-2 spectral regions. This revealed that the green band (540–578 nm) was most important for discriminating between trees and lianas (Figure S1, Supplementary Materials).

We also calculated a variety of vegetation indices to assess whether one simple metric is capable of discriminating between liana-free and liana-infested pixels. As the green band was the most effective, we specifically calculated indices that may promote signals in the green spectrum such as, Greenness Index (GI) which has shown to outperform other indices when discriminating vegetation using the visible spectra [48]. We assessed which vegetation index was most effective at separating severely (>75%) and low (<25%) liana

infested pixels by comparing their effect size (Table S1). We used Cliff's delta, which is considered to be a robust measure of effect size, to calculate the magnitude of difference between the two groups [49]. Cliff's delta computes the probability that a randomly selected observation from one group is larger than an observation from another group:

$$\sum \frac{[x > y] - [x < y]}{mn} \tag{1}$$

whereby *x* and *y* are liana-free and severely liana infested pixels and *m* and *n* are the number of pixels within each group.

To assess whether a single vegetation index could be used to detect liana infestation in satellite imagery over broad spatial scales we tested whether the vegetation index varied in response to forest type (i.e., primary and selectively logged forests) as well as during and after a period of El Niño-induced drought (i.e., across years). Subsequently, we used a linear regression model with an interaction term to allow the effect of airborne-derived liana infestation on the vegetation index to vary by forest type or year. Liana infestation classes were defined based on the separability of spectra for each group [28,29]. We also used a linear mixed effects model to account for variation in forest type. We tested whether the relationship between the vegetation index and liana infestation differed across the four years by using a pairwise comparison of the slope coefficients.

### 2.3.3. Modelling Liana Infestation

To predict liana infestation in satellite imagery we used a neural network classification trained using the airborne liana infestation output. To accurately predict liana infestation in satellite-based imagery as well as to test the efficacy of a single vegetation index we modelled liana infestation using three different sets of input variables: (1) vegetation index only, (2) all Sentinel 2-bands and (3) all Sentinel 2-bands and the vegetation index.

The same model construction and process was applied (as in the airborne-derived liana infestation assessment). As the spatial resolutions of the satellite (10 m) and airborne (3 m) imagery do not match, we degraded the resolution of the airborne imagery so both products had a resolution of 10 m. We used pixels from the airborne-derived liana infestation output classified as having no infestation or completely liana infested as training data. Values greater than 95% were therefore classified as a 'liana' and values less than 5% were classified as a 'tree'. This yielded a total of 3622 pixels with no (<5%) liana infestation and 6128 pixels completely (>95%) liana infested. Data were balanced to ensure there was an equal number of data points within each input class (i.e., 3622 pixels). Data were split 80% for training and 20% for testing. We ran the neural network model 100 times and after each iteration the model was applied to the entire study landscape. We propagated error associated with uncertainty in the airborne liana infestation assessment by using each of the 100 airborne derived liana infestation outputs to train the satellite-based models. With each iteration we repeated the following steps; (1) removed pixels that were completely liana infested to ensure each input class was balanced and (2) split data for training and testing. A final satellite-derived liana infestation map was calculated by averaging all of the 100 neural network outputs.

### 2.3.4. Temporal Change in Liana Infestation

To reliably assess a degree of change in predicted liana infestation over time, we focused on change between low [<25%] and severe [>75%] liana infestation classes within the primary forest. Individual neural network models were trained for each of the four years. The percentage of pixels classified as having either low or severe liana infestation were calculated for each year to indicate a level of change over time. Additionally, we calculated the percentage of pixels that changed from low to severe and vice versa from 2016 to 2019. This process was repeated for each of the 100 satellite-derived liana infestation outputs. This allowed for a calculation of uncertainty around estimates of change in liana infestation over the four years.

### 2.3.5. Accuracy Assessment

To assess the accuracy of the satellite-derived liana infestation output we used a random selection of pixels (n = 200) from both the airborne- and satellite-derived liana infestation outputs. This process was repeated 10 times and the relationship between the predictions were assessed after each iteration. To assess the accuracy of the predicted liana infestation outputs we estimated the root mean squared deviation (RMSD) as:

$$\sqrt{\frac{1}{n-1}\sum_{i=1}^{n}(\hat{y}_i - y_i)^2} \qquad (2)$$

which represents the mean deviation of predicted $\hat{y}_i$ from observed values $y_i$ (i.e., with respect to the 1:1 line) [50]. We also calculated the relative bias, expressed as a percentage:

$$\frac{1}{n}\sum_{i=1}^{n}\frac{LI_{pred} - LI_{obs}}{LI_{obs}} * 100 \qquad (3)$$

whereby, $LI_{obs}$ and $LI_{pred}$ denote observed and predicted liana infestation. This gives an indication of the degree to which satellite-derived liana infestation may be over- or under-predicted relative to airborne-derived liana infestation.

To increase the level of confidence around estimates of liana infestation we also degraded outputs to an ordinal scale by partitioning predicted liana infestation into four groups as follows: neural network membership values equal to or below 0.25 were set to 'low', values between 0.26 and 0.50 were set to 'medium', values between 0.51 and 0.74 were set to 'high' and values equal to or greater than 0.75 were set to 'severe'. To assess the accuracy of liana infestation classes, we produced a confusion matrix using satellite-derived liana infestation (predicted) and airborne-derived liana infestation (observed) and calculated the overall accuracy, specificity, sensitivity, balanced accuracy and area under the curve (AUC).

## 3. Results

Satellite-based spectral reflectance in the visible spectrum, and predominantly in the green reflectance region, was most effective at separating low (<25%) and severe liana infestation (>75%) classes derived from airborne-hyperspectral data (Figure S1). Subsequently, we found Greenness Index (GI) to be the most effective vegetation index for discriminating between low and severe liana infestation (Table S1). We also found that average predicted greenness values derived from satellite imagery increased significantly in response to an increase in liana infestation and were significantly greater in the logged forest in comparison to the primary forest (Figure 2a, Figure S2). In addition, average predicted greenness values were positively related to liana infestation in all four years (Figure 2b). However, there was a greater increase in greenness relative to an increase in liana infestation in 2016 (drought year) in comparison to other years, as shown by significant differences in slope coefficients (Q1) (Table 1). Slopes did not differ between the three non-drought years, except for a weak significant difference between 2017 and 2018 (Table 1).

**Table 1.** Pairwise comparison of linear regression slope coefficients (in Figure 2b). *p*-values adjusted using Bonferroni correction. Significance level set at 0.05.

| Contrast (Years) | Estimate | SE | df | t | *p* |
|:---:|:---:|:---:|:---:|:---:|:---:|
| 2016–2017 | 0.01583 | 0.00158 | 34852 | 10.038 | <0.0001 |
| 2016–2018 | 0.01170 | 0.00158 | 34852 | 7.415 | <0.0001 |
| 2016–2019 | 0.01300 | 0.00158 | 34852 | 8.240 | <0.0001 |
| 2017–2018 | −0.00414 | 0.00158 | 34852 | −2.623 | 0.0524 |
| 2017–2019 | −0.00284 | 0.00158 | 34852 | −1.798 | 0.4330 |
| 2018–2019 | 0.00130 | 0.00158 | 34852 | 0.825 | 1.0000 |

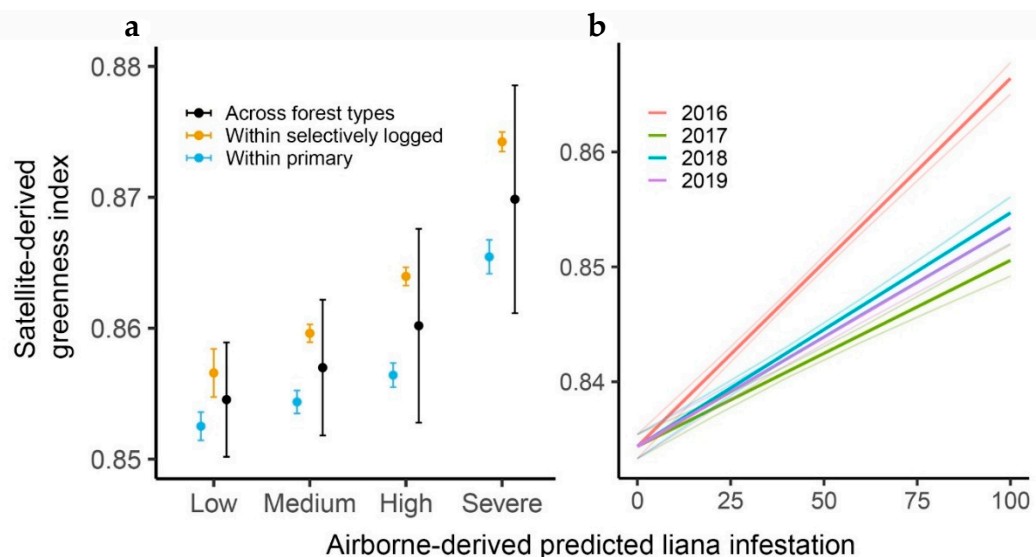

**Figure 2.** Satellite-derived predicted mean greenness (**a**) combined for all years (2016–2019) across forest types (black) and within the primary (blue) and selectively logged (orange) forests in response to airborne-derived liana infestation classes and (**b**) for each year in response to airborne-derived liana infestation percent cover. Liana infestation classes are defined as: Low [<25%], Medium [26–50%], High [51–74%] and Severe [>75%]. Error bars are 95% confidence intervals based on error in model fit as well as uncertainty derived from error in airborne-derived liana infestation estimates.

A neural network classification using GI as the only input variable was not capable of accurately predicting liana infestation in satellite-based imagery (Q2) (Figure 3a,d). While predicted mean greenness values showed a clear increasing trend in response to an increase in liana infestation (Figure 2a,b), large variation in greenness values ultimately limited its use as a single predictor variable (Figure 3a,d, Table S2). Using all Sentinel-2 bands without GI increased the accuracy of satellite-based predictions in the primary (AUC: 0.76) and logged (AUC: 0.7) forests (Figure 3b,e, Table S2). Furthermore, combining all Sentinel-2 bands and GI provided a further increase in accuracy within the primary (AUC: 0.8) and logged (AUC: 0.71) forests (Q3) (Figure 3c,f, Table S2). In addition, we found a larger underestimation of satellite-derived liana infestation, relative to liana infestation obtained from airborne data, in the logged forest (bias = $-15.5$ and $-14.8$) in comparison to the primary forest (bias = $-9.5$ and $-6.2$) for the model using only Sentinel-2 bands and the model using Sentinel-2 bands and GI, respectively.

To assess change in liana infestation over time we used the output from the model using Sentinel-2 bands and GI which revealed the greatest accuracy (AUC: 0.99) (Table S2). The percentage of pixels classified as severe liana infestation showed a sustained and significant increase over time, from 12.9% ± 0.63 (95% CI) in 2016 to 17.3% ± 2 in 2019 (Q4) (Figure 4c, Table S2). However, the low liana infested pixels did not show a similarly consistent downward trend and instead remained more or less constant over the three-year period (35.4% ± 3.6 in 2016 to 33.6% ± 3.2 in 2019). Liana infestation at a pixel level was dynamic, with 2.66% ± 0.76 of pixels having changed from low to severe and 1.22% ± 0.2 having changed from severe to low liana infestation from May 2016 to April 2019. Taken together, these results indicate a potential forest-wide increase in severe liana infestation.

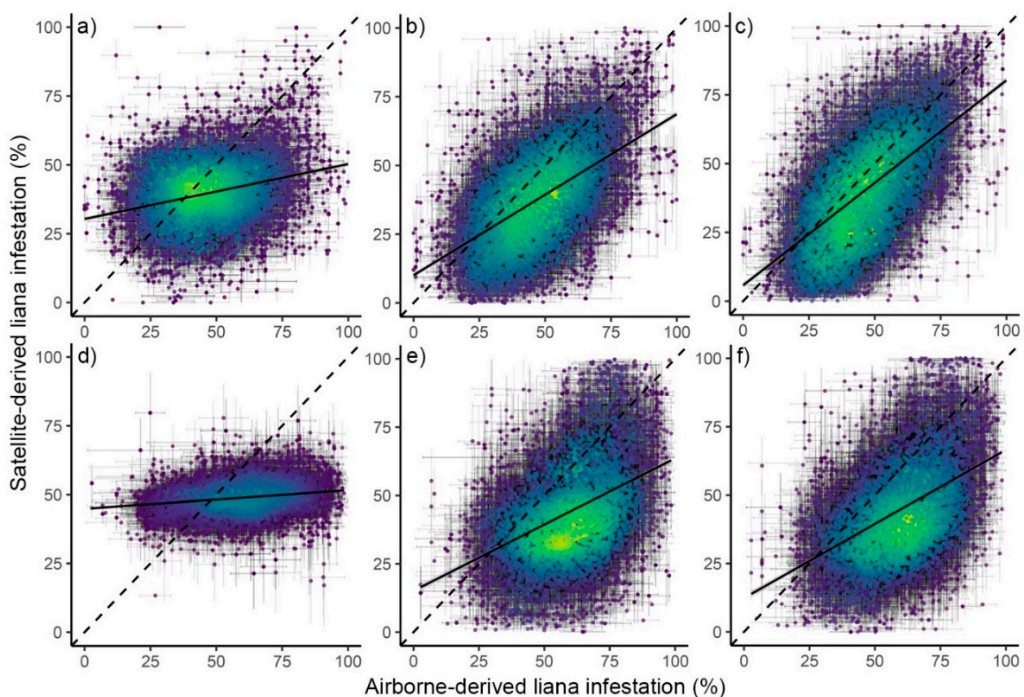

**Figure 3.** Relationship between airborne- and satellite-derived liana infestation with input variables consisting of (**a**) Greenness Index, (**b**) all Sentinel-2 bands and (**c**) Greenness Index and all Sentinel-2 bands in the primary forest and (**d**) Greenness Index, (**e**) all Sentinel-2 bands and (**f**) Greenness Index and all Sentinel-2 bands in the selectively logged forest. Dashed lines represent a 1:1 lines. Solid black lines correspond to linear models. Colored points correspond to the density of overlapping points ranging from purple to yellow with increasing density. Error bars represent ± 1 standard deviation.

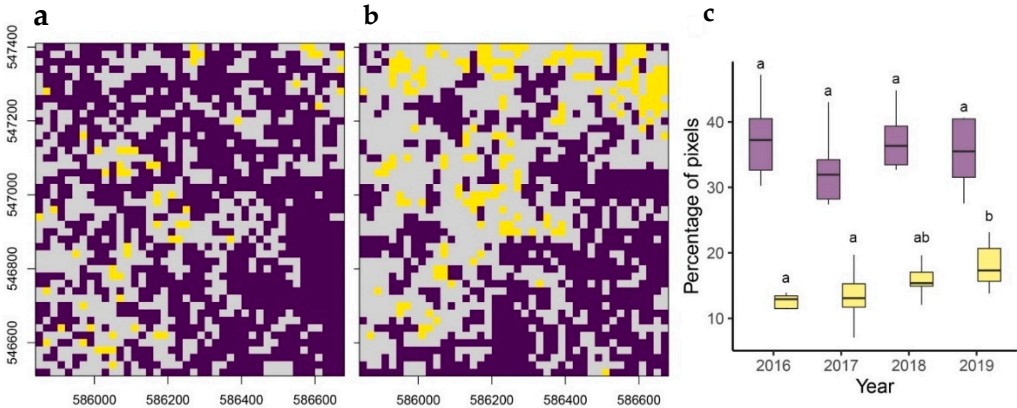

**Figure 4.** Predicted liana infestation in two classes, Low [≤25%] (purple) and Severe [≥75%] (yellow), derived from Sentinel-2 satellite imagery showing an extract from (**a**) 2016, (**b**) 2019. Grey areas correspond to liana infestation values not within low or severe classes (i.e., 26–74%). Panel (**c**) the percentage of pixels within each class for all four years (2016–2019). Letters in (**c**) indicate statistically significant differences between years as assessed using a least significant difference test with Bonferroni adjusted *p*-values. Significance level was set at 0.05.

## 4. Discussion

This study provides evidence, for the first time, that liana infestation can be detected in a closed-canopy tropical forest using multispectral satellite-based imagery. Furthermore, satellite-derived Greenness Index showed clear separation in response to airborne-derived liana infestation classes within both primary and selectively logged forests as well as during periods of wet and dry conditions (Figure 2). These results indicate that reflectance

in the visible spectra (546–574 nm) was most efficient in distinguishing lianas from trees (Figure S1) and in particular Greenness Index (GI) was found to be an effective metric (Table S1). This corroborates results from other studies that found the visible spectral region to be the most important for spectrally discriminating between lianas and trees [18,28,37]. Previous studies have shown higher reflectance of liana leaves in the visible region consistent with lower levels of chlorophyll content in lianas than in trees [18,37]. Subsequently, an increase in liana canopy cover will result in higher values of greenness.

A significant positive relationship between GI and liana infestation was found across all four years for which imagery was obtained, however there was a greater increase in greenness relative to an increase in liana infestation for the year in which the El Niño-induced drought occurred (Q1) (Figure 2b). A greater increase in greenness in the drought year may be attributed to: (1) a reduction in tree greenness (i.e., at 0% liana infestation), (2) an increase in liana greenness or (3) a combination of both. Lianas generally seem to experience less water stress due to their ability to access and use water more efficiently than co-occurring trees during seasonal or periodic droughts [51–53]. Evidence from dry forests show that the chlorophyll concentration of liana leaves is lower than for trees, and this difference results in an increase in reflectance in the visible spectra [37]. However, in wetter forests chlorophyll concentration is observed to simultaneously increase in liana leaves and decrease in tree leaves [37], which leads to a lower spectral contrast between trees and lianas.

While one single vegetation index was not capable of predicting liana infestation (Q2), a strong relationship between greenness and liana infestation was found. Furthermore, high accuracy was obtained when using all Sentinel-2 bands in combination with a neural network classification to predict liana infestation at the landscape-level (Q3). This indicates that interpretation of forest-wide responses to environmental or climatic changes using satellite imagery may be problematic if lianas are interpreted as tree canopies. Lianas are a particularly dominant and wide-spread feature of tropical canopies [7,54] and therefore their presence may obscure or distort satellite-derived spectral reflectance of tree canopies. Furthermore, as the effect of increased liana infestation on greenness differed under different climatic conditions (Figure 2b), satellite-observed changes in spectral reflectance in response to climatic changes, e.g., [55] may be complicated cf. [56] by the differential responses of lianas and trees. This highlights the importance of accounting for the effect of liana infestation on satellite-derived reflectance metrics to ensure the accurate interpretation of remotely sensed multispectral data, especially given evidence of increasing liana biomass and abundance.

We found that temporal changes in liana infestation can be observed using a time-series of satellite-based imagery (Q4). Severely liana-infested pixels ($\geq$75% infested by lianas) increased significantly over time from 12.9% $\pm$ 0.63 in May 2016 to 17.3% $\pm$ 2 in April 2019, whilst low ($\leq$25%) liana infestation remained relatively constant 35.4% $\pm$ 3.6 to 33.6% $\pm$ 3.2 over the same three-year period (Figure 4). This degree of change is minimal compared to change in seasonal forests, where lianas show more rapid growth and have a significant growth advantage over trees compared with aseasonal forests [57]. For example, in a seasonal forest, an increase of 65% of trees with severe liana infestation was observed over a 10-year period [7] in contrast to an increase of around 3% of trees with severe liana infestation in Peninsula Malaysia over the same time period [58]. In this study we found an increase of 4.4% over three years, however low liana infestation remained relatively constant over the same time period. This suggests that increases are limited to severe liana infestation which may not necessarily represent an increase in the overall percentage of infested trees. Furthermore, assessing the level of change in the severe liana infestation class over time is subject to error. While the accuracy of predicted liana infestation in low ($\leq$25%) and severe ($\geq$75%) classes revealed good accuracy (AUC: 0.99), accuracy was reduced when liana infestation was predicted across all infestation classes (Figure 3; Table S2). Subsequently, error in the classification of severe liana infestation, due to misclassification of pixels in the high (50–74%) liana infestation class, may have

led to an under- or over-estimation of change in liana infestation over time. Even though there is no ground data to support this, these results imply that, despite the proportion of low liana-infested pixels remaining relatively constant, severely liana-infested pixels may have increased by 4.4% over the three-year time period. If indeed true, this suggests that an increase in liana abundance may not be confined to the Neotropics, as indicated by previous studies, e.g., [11,58]. A possible driver of the increase in liana infestation may be that lianas tend to favor dry conditions and exhibit a dry season growth advantage over trees [57]. Whilst Borneo has one of the most aseasonal climates of any tropical region [41] (Whitmore 1984), recent evidence has suggested that Borneo may be experiencing hotter and drier conditions driven by continued deforestation [42,59,60], which is likely to provide favorable conditions for liana growth [57,61].

However, similar to studies that assessed individual tree crowns [7,58], these results also indicate that liana infestation is dynamic, with 2.66% ± 0.76 of pixels changing from low to severe and 1.22% ± 0.2 changing from severe to low liana infestation over the three-year time period. This represents a total of 3.9% change between classes over a 3-year period. Available data from aseasonal and seasonal forests show that changes between low and severe liana infestation classes over time scales longer than three years range from 4–16.2% [7,58]. A possible explanation for this relatively high change in liana infestation could be related to the El Niño-induced drought which occurred in 2016 [44]. During this time, tree growth may have temporarily slowed and lianas may have had a growth advantage which may have resulted in more dynamic changes in liana infestation over this period. A longer time-series is therefore needed to provide conclusive results of whether there is a temporal increase in liana infestation and how this may impact on the ability of these forests to store and sequester carbon.

Evidence that lianas can be detected across closed-canopy forest using satellite imagery provides a substantial advance in our ability to monitor change in liana infestation over time. Furthermore, evidence of this relationship under different climatic conditions and across forest types suggests that this methodology should apply broadly. However, there are a number of limitations to the current study. First, satellite-based liana infestation predictions on a continuous scale seemed to underestimate liana infestation compared to airborne predictions. Even though there was a small bias in predictions in the primary forests, underprediction was a particular problem in the selectively logged forests. However, a high classification accuracy (0.88) for predicted liana infestation for low (≤25%) and severe (≥75%) classes in the selectively logged forest was found (Table S2). Therefore, prediction within classes may be required in order to compare liana infestation between forest types.

Second, satellite-based images were trained with the same airborne-derived liana infestation output, to assess temporal changes in liana infestation over time. This may be problematic given the dynamic nature of liana infestation [7,58]. For example, changes across the landscape, such as the formation of canopy gaps, may have led to changes in liana infestation over time which are not reflected in the training data. This would have led to certain areas across the landscape being trained incorrectly, and therefore may result in an increase in error around liana infestation predictions over time. However, as change was assessed over a relatively short time period, it is unlikely that this would have affected a large area of the forest. Furthermore, a small degree of error in training data has shown to have little impact on the accuracy of predictions [29] and therefore it is unlikely that these results are severely confounded by using the same training data.

Third, the level of exposure to sun light may affect spectral reflectance, which, in turn, may make it more difficult to detect liana infestation. For example, large canopy gaps will be more exposed to light whereas smaller canopy gaps and some tree crowns may be affected by shadow from nearby tall trees. While the effect of shadow has shown to impact some vegetation indices, the effect has shown to be less on NDVI and Greenness Index [62]. Therefore, while Greenness Index, as a sole input variable, was unable to accurately predict liana infestation (Figure 3a,d), it is possible that including Greenness Index assists in the detection of liana infestation in areas affected by shadow. Indeed, the inclusion of

Sentinel-2 bands which cover the NIR- and SWIR- regions appear to be essential in order to discriminate between trees and lianas [20,21]. This also suggests that the detection of liana infestation should be achievable across a variety of multispectral sensors.

Lastly, while this methodology appears to provide an accurate assessment of liana infestation for the region to which it was trained, it may be limited in its broad applicability across forests in different regions. The accuracy of satellite-derived liana infestation is obtained relative to airborne-derived predictions. However, this represents the same area in which the model was trained upon. It is therefore likely that there could be a reduction in classification accuracy for areas outside the training extent. The use of this current model with no additional training data may therefore only be accurately applied to nearby areas and may require a classification to be restricted to classes of low and high/severe liana infestation if being applied further afield. Future work should consider the feasibility to obtain a generalized model to predict liana infestation over regional- or continental-scales.

In summary, this work has demonstrated that during dry periods, or drought events, lianas and trees are more spectrally distinct (Q1). While one vegetation index (Greenness Index) is not capable of accurately predicting liana infestation (Q2), the use of all spectral bands in combination with a neural network classification was capable of predicting liana infestation with a high degree of accuracy (Q3). Lastly, temporal changes in liana infestation can be observed using a time-series of satellite-based imagery; however, only change between low [<25%] and severe [>75%] liana infestation classes was possible due to increased mis-classification in medium [25–75%] infestation classes.

## 5. Conclusions

We have shown, for the first time, that satellite-based imagery can be used to accurately predict liana infestation during both wet and drought years and across forest types, which suggests this methodology should apply broadly. The use of satellite remote sensing therefore advances our ability to assess the distribution of liana infestation over time and across forests globally. This in turn will assist in providing insights into the drivers responsible for the distribution and change in liana infestation at multiple spatial and temporal scales as well as quantify the liana-induced impacts on carbon dynamics in tropical forests.

**Supplementary Materials:** The following are available online at https://www.mdpi.com/article/10.3390/rs13142774/s1, Figure S1: Airborne-derived standardized ($\mu$ = 0, $\sigma$ = 1) hyperspectral reflectance of liana-free trees and trees severely infested with liana leaves; Table S1. Comparison of vegetation indices for the seperation of low ($\leq$25%) and severe ($\geq$75%) liana infestation classes within the primary and logged forests and across the full landscape for each of the four satellite-derived images (2016–2019); Table S2. Accuracy assessment for predicted liana infestation in satellite-based multispectral imagery using three different sets of input variables: (1) all Sentinel 2-bands, (2) Greenness Index (3) all Sentinel 2-bands and Greenness Index within the primary and selectively logged forests; Figure S2. Difference in greenness between the primary and logged forest for imagery collected from 2016 to 2019.

**Author Contributions:** C.J.C., G.M.F.v.d.H., D.S.B. and G.M.F. designed the research. C.J.C. collected ground-derived liana canopy cover data. C.J.C. performed the data analysis and led the writing of the manuscript. All authors contributed critically to the draft manuscripts and gave final approval for publication. Conceptualization, C.J.C., G.M.F.v.d.H., D.S.B. and G.M.F.; formal analysis, C.J.C.; funding acquisition, G.M.F.v.d.H., D.S.B. and G.M.F.; methodology, C.J.C.; writing—original draft, C.J.C.; writing—review and editing, G.M.F.v.d.H., D.S.B. and G.M.F. All authors have read and agreed to the published version of the manuscript.

**Funding:** The authors thank the Natural Environment Research Council [NE/P004806/1 to MEJC, D.S.B., G.M.F., G.M.F.v.d.H.; NE/I528477/1 (ARSF MA14/11) to MEJC, D.S.B., G.M.F. and NE/L002604/1 to D.S.B., G.M.F., G.M.F.v.d.H.;] as well as the University of Nottingham for an Anne McLaren Research Fellowship to G.M.F.v.d.H. which funded the collection of the ground data.

**Data Availability Statement:** All the data relating to assessing the spatial distribution of liana infestation are accessible on the University of Nottingham Research Data Repository, doi:10.17639/nott.7092.

**Acknowledgments:** The authors thank all the field assistants and staff at Danum Valley as well as supporting agencies including Sabah Biodiversity Center, Danum Valley Management Committee, Sabah Forestry Department and the Chief Minister's Department Office of Internal Affairs & Research for providing logistical support.

**Conflicts of Interest:** The authors declare no conflict of interest.

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
