# Peer review of "Detection of Spatial and Temporal Patterns of Liana Infestation Using Satellite-Derived Imagery"

_remotesensing, doi:10.3390/rs13142774_

Round 1
Reviewer 1 Report
The study by Chandler and colleagues shows that satellite-based imagery can be used to assess temporal and spatial patterns of liana infestation in aseasonal tropical forests of Sabah, Malaysia. What is more important, however, is that the presented methodology seems to be capable of detecting a forest-wide increase of liana infestation in response to an ENSO drought period. If this pattern can be observed across other tropical regions as well, such analysis could be used to quantify liana induced impact on tropical carbon dynamics in response to projected increasing drought periods. In general, the study findings are very well presented and need no further clarification. Nonetheless, I would suggest to consider the following minor comments indicated below.
Could the text structure still be improved by including hypothesis or research questions at the end of the introduction section?
Something that strikes me is the differentiation between the low and severe infestation classes in Figure 4. Could you explain as to why there appears to be a clear gradient between these classes in the pixels spreading from the upper left to the lower right corner of the figure?
L36: is stronger “than”
L159: and an “output” layer?
L401: do you mean “tree growth may have been temporarily slowed down”?
L405: to store and “sequester” carbon.
L429-440: The clarity of arguments presented in this section could be improved by moving up this paragraph before L418 (i.e., “third” before “second”).
L455: which suggests “that” this methodology should apply broadly.
Reviewer 2 Report
The study explored the indentification of lianas and temporal changes in liana infestation in closed-canopy tropical forest using satellite imagery and neural network classification. The various VIs were tested and response to drought was analysed in liana infestation periods for four years.
Comments to Authors are provided below:
- Stusy area: as study tested a response to drought, some information about drought in study area in 2016, 2017, 2018, 2019 is needed. As annual presipitation, temperature, solar radiation..
- Matherials and Methods. It is recommended to split data description and methods of data processing used to increase the transfarability of the MS structure. As a suggestion: 2.2. Data, 2.2.1. Airborne data, 2.2.2.Satellite data, 2.2.3. Field data. 2.3. Identification of liana infestation. 2.3.1. from airborne data, 2.3.2. from satellite data. 2.4. Accuracy assessment.
- line 192 - green band - please, add a length of the band,
- line 287 - liana infestation classes were assigned based on Authors experience or forest inventory report? reference on the source.
- Discussion: it is recommended to split the text into paragraphs corresponded to the study aimed - detection of lianas using neural network classification, analysis of VIs, responcse to drought, temporal chnges of liana infestation.
Reviewer 3 Report
The goal of this work is listed to determine the efficacy of satellite-based remote sensing for the detection of liana infestation across an aseasonal tropical forest in Sabah, Borneo; and to also assess the detectability of liana within primary and selectively logged forests as well as related to a period of drought. Specifically, the authors aimed to test whether 1) liana infestation can be detected in satellite-based imagery using a neural network classification trained by airborne-derived liana infestation, 2) one single vegetation index is capable of detecting liana infestation, 3) a response to drought facilitates the differentiation in spectral reflectance for lianas versus trees and 4) temporal changes in liana infestation can be observed using a time-series of satellite-based imagery.
The manuscript is well-written with solid citations. The one step that creates uncertainty about the work is the liana canopy cover data were collected over a three-year period, and those years did not match the year the airborne hyperspectral data was collected, and then the Sentinel data is another year. Techniques are used to try to deal with these mismatches, including a reference cited.
Although it appears the study addresses the four tests listed in the goal and tests listed in the paragraph above, the information is not in the same order as the tests, and not written to directly address the hypotheses. What would really help is to address the four hypotheses (L113-118) at the end of the discussion (L406-451) directly. Clearly repeat that the neural net classification did not work well. For #2, clearly state that one single vegetation index was not found to be robust in detecting liana infestation. Etc. For #4, clearly state that temporal changes in liana infestation were observed using a time series, though there is uncertainty due to imprecision in the field data. (if I have mis-stated anything, it is because this is not written clearly in the current manuscript.)
Please make the responses to the tests clear.
Round 2
Reviewer 2 Report
I appreciate the authors effort. The manuscript has been improved. Thank you for the answers.